# Inheritance of Some Traits in Crosses between Hybrid Tea Roses and Old Garden Roses

**DOI:** 10.3390/plants13131797

**Published:** 2024-06-28

**Authors:** Tuğba Kılıç, Soner Kazaz, Ezgi Doğan Meral, Emine Kırbay

**Affiliations:** 1Horticulture Department, Yozgat Bozok University, Yozgat 66200, Türkiye; 2Horticulture Department, Ankara University, Ankara 06110, Türkiye; soner.kazaz@ankara.edu.tr; 3Horticulture Department, Bingöl University, Bingöl 12000, Türkiye; ezgidgn23@gmail.com; 4Ataturk Health Services Vocational School, Afyonkarahisar Health Sciences University, Afyonkarahisar 03030, Türkiye; eminekirbay86@gmail.com

**Keywords:** rose, hybridization, quantitative characters, scent, recurrent blooming, narrow-sense heritability

## Abstract

The limited knowledge about the inheritance of traits in roses makes the efficient development of rose varieties challenging. In order to achieve breeding goals, the inheritance of traits needs to be explored. Additionally, for the inheritance of a trait like scent, which remains a mystery, it is crucial to know the success of parental traits in transmitting them to the next generation. Understanding this allows for accurate parental selection, ensuring sustainability in meeting market demand and providing convenience to breeders. The aim of this study was to assess the success of cross-combinations between scented old garden roses and hybrid tea roses used in cut roses in transferring their existing traits, with the objective of achieving scented cut roses. The evaluated traits included recurrent blooming, flower stem length, flower diameter, petal number, scent, and bud length of both parents and progenies. The inheritance of these traits was evaluated through theoretical evaluations, including calculating heterosis and heterobeltiosis and determining narrow-sense heritability. The combinations and examined traits were assessed using a hierarchical clustering heat map. The results of this study indicated that flower stem length, flower diameter, petal number, and bud length traits had a moderate degree of narrow-sense heritability, suggesting the influence of non-additive genes on these traits. This study observed a low success rate in obtaining progenies with scent in cross combinations between cut roses and old garden roses, indicating the challenges in obtaining scented genotypes. The discrepancy between the observed phenotypic rates and the expected phenotypic and genotypic rates, according to Punnett squares, suggests that the examined traits could be controlled by polygenic genes. The progenies were observed to exhibit a greater resemblance to old garden roses than hybrid tea roses and did not meet the commercial quality standards for cut flowers. The significant negative heterosis observed in 65.12% (petal number) and 99.61% (flower diameter) of the progenies provides strong evidence of resemblance to old garden roses. Considering these findings, it is recommended to consider old garden roses as parents, taking into account their suitability for other breeding objectives.

## 1. Introduction

The rose, which belongs to the *Rosa* genus in the Rosaceae family, is naturally distributed in the Northern Hemisphere, including Asia, Europe, the Middle East, and North America. It is one of the plant species widely used in the ornamental plant industry, cosmetic industry, food, and medicine sectors [1]. There are more than 100 to 250 species of roses identified worldwide [2]. The haploid chromosome number of roses is *x* = 7, and chromosome numbers vary depending on the ploidy level, ranging from *2n* = *2x* = 14 in diploids to *2n* = *8x* = 56 in octoploids. In a recent study, an endemic wild rose species (*Rosa praelucens* Byhouwer) with a chromosome number of *2n* = *10x* = 70 has also been reported [3].

The rose is the most traded cut flower in the world [4]. Through breeding studies, thousands of new varieties have been developed to meet consumer expectations. More than 37,000 rose varieties have been introduced to the sector [5]. Crossbreeding is preferred in the development of new rose varieties because it generates greater genetic variation and leads to improved outcomes [6]. The cost is also lower compared to the use of molecular methods [7].

In the breeding of cut roses, certain characteristics such as flower stem length and thickness, bud size, yield, disease tolerance, flowering time, and vase life gained prominence [8,9]. However, consumer preferences evolve over time, and there has been a growing interest in scented cut roses in recent years. Although the natural scent of roses is considered economically significant, the scent has been lost in cut roses due to intensive breeding efforts focused on other traits [10]. Most commercial cut rose varieties have no scent since the scent was not among the desired selection criteria for many years [11].

Due to the increase in consumer demand for scented cut roses, breeding companies have planned programs to incorporate the scent trait into cut roses by including scent as one of the selection criteria [11,12]. Although a limited number of scented cut rose varieties have been developed, rose breeding is primarily carried out by highly competitive commercial companies, resulting in the genetic control of traits being treated as proprietary information [13,14]. Additionally, there is still a lack of sufficient scientific literature on the breeding of scented cut roses that meet commercial quality standards, and the inheritance of these traits has not yet been fully elucidated [10].

Most of the old garden and wild rose species that have spread worldwide are scented, such as *R. alba* L., *R. damascena* Mill., *R. centifolia* L., *R. odorata* L. cv. Louis XIV, *R. gallica* L., and *R. moschata* Herrm [15,16,17]. However, these rose species do not have commercial use in the cut flower industry because their morphological characteristics do not meet the criteria for commercial quality. Moreover, it is not clearly known how successful old garden roses are in developing varieties that meet the cut flower quality criteria. The lack of sufficient information on breeding scented and commercial-quality cut roses poses a significant challenge for researchers and amateur breeders. To develop new scented varieties suitable for commercial quality criteria in cut roses, it is essential to understand the ability to transfer the characteristics of these species to the next generation and have information about the inheritance of the desired traits [18].

It is believed that the rate of developing new varieties that are scented and meet commercial quality criteria may be high through crosses between old garden roses and commercial cut rose varieties. This study aimed to assess the ability of old garden roses to transfer certain plant and flower characteristics to the next generation and to gather information about the inheritance of these traits through crosses between cut roses and old garden roses. The results of this study are expected to aid in identifying suitable materials for breeding new high-quality scented cut flower varieties.

## 2. Results and Discussion 

### 2.1. Qualitative and Quantitative Traits of F_1_ Progenies

In the combinations where Damask rose was used as the pollen parent, an evaluation of the traits could not be made since neither the progeny could be obtained (Avalanche × Damask rose and Sweet Avalanche × Damask rose) nor was recurrent blooming observed, and there was a long period of juvenile sterility (Layla × Damask rose, Samourai × Damask rose, Magnum × Damask rose, and First Red × Damask rose). Therefore, measurements were conducted on 258 progenies.

### 2.2. Recurrent Blooming

Among all F_1_ progenies, 83.06% showed recurrent blooming. In all cross combinations in which the Black rose was used as the pollen parent, all genotypes had the recurrent blooming trait. Some F_1_ progenies of cross combinations in which Damask rose was used as a pollen parent did not show recurrent blooming. Moreover, some of them did not show any blooms for a year. Juvenile sterility has been observed. In the F_1_ progenies of cross combinations in which Cabbage rose was used as a pollen parent, the rate of progenies showing recurrent blooming depending on the seed parent, except Sweet Avalanche, varied between 20% and 100% (Figure 1). 

In previous studies conducted to investigate the inheritance of the recurrent blooming trait in roses, it was reported that recurrent blooming was controlled by a homozygous recessive gene [19,20,21]. However, Shupert [22] found that the observed rates of recurrent blooming did not align with the expected rates, challenging the hypothesis of a homozygous recessive gene controlling the trait. Jones [14] determined that among 11 different cross combinations, one combination deviated from the expected rate, with several genotypes blooming only once a year. Shubin et al. [23] discovered that none of the 296 hybrid progenies obtained from cross combinations between *R. chinensis* ‘Old Blush’ (recurrent blooming) and *R. wichuriana* Basye’s Thornless (blooming once a year) exhibited recurrent blooming. Additionally, they reported that out of 300 hybrid progenies obtained from backcrossing F_1_ progenies with *R. chinensis* ‘Old Blush,’ only 83 displayed recurrent blooming, which was inconsistent with the expected rate. 

The results of this study are in line with the findings reported by Semeniuk [19,20], de Vries and Dubois [24], and Debener [21], suggesting that in combinations using Black rose and Damask rose as pollen parents, the recurrent blooming is controlled by a homozygous recessive gene. All progenies obtained from crosses between recurrently blooming Black roses and recurrently blooming hybrid tea roses exhibited recurrent blooming. Similarly, the progenies obtained from crosses between once-blooming Damask roses and recurrently blooming hybrid tea roses bloomed once a year. When assuming that recurrent blooming is controlled by a homozygous recessive gene, the scenarios of hybrid tea roses, Black roses being homozygous recessive for the recurrent blooming trait, and Damask roses being homozygous dominant are consistent.

It is expected that all progenies should display recurrent blooming in crosses between recurrent blooming hybrid tea roses and Cabbage roses, where the Cabbage rose is presumed to be homozygous recessive, similar to the Black rose, for exhibiting recurrent blooming. However, both recurrent blooming and once-blooming progenies were observed. When Cabbage rose was assumed to be heterozygous, whether hybrid tea roses were considered heterozygous or homozygous, the segregation ratios did not match the expected ratios. These findings are not in line with the notion that the recurrent blooming trait is controlled by a single gene. For instance, in the case where both parents are assumed to be heterozygous, the number of progenies not exhibiting recurrent blooming in the Samourai × Cabbage rose combination was much higher than expected. Similar results have been reported by Jones [14], where the number of recurrent blooming progenies was found to be less than the number of non-recurrent blooming progenies.

Based on these findings, the recurrent blooming trait may be controlled by multiple genes. There are other studies suggesting that the recurrent blooming trait in diploid roses is controlled by two recessive genes, as reported by Shubin et al. [23] and Smulders et al. [25].

Another significant finding of this study is the occurrence of juvenile sterility in progenies resulting from crosses involving Damask rose. According to Zlesak [26], juvenile sterility in once-year-blooming roses can persist for one to several years. It is possible that the traits of recurrent blooming and juvenile sterility are influenced by the same genetic mechanisms.

### 2.3. Scent

The highest rate of the scented progenies appears to have been determined in the Samourai × Cabbage rose and Sweet Avalanche × Cabbage rose combinations. However, only one F_1_ progeny survived in both cross combinations. Since an adequate number of hybrid progenies could not be obtained to demonstrate segregation, neither of the cross combinations was evaluated individually. Among the F_1_ progenies, 78.93% were determined to be scentless or had a barely perceptible scent, whereas 21.07% were scented. Among the scented hybrid progenies, only 5.66% were identified as having a strong scent. These results were obtained when Layla and Sweet Avalanche were used as seed parents (Figure 2).

In studies on the inheritance of scent in roses, it has been reported that scent is a homozygous recessive trait controlled by polygenic genes [27,28]. In a hybridization study conducted by Cherri-Martin et al. [11] using two different rose varieties known for their distinct scent compositions, it was found that the concentration and diversity of volatile compounds in the hybrid progenies were lower compared to those in the parents. The researchers observed that while the parents contained 30 different volatile compounds, only 11 of these compounds were in the progenies. Furthermore, they observed that the variation in scent quality correlated with the levels of monoterpenes, indicating the influence of these compounds on the overall scent characteristics, and that the hybrid genotypes possessed scent metabolism characteristics from both parent plants. The researchers reported a limited occurrence of hybrid progenies with a pleasant scent, suggesting the complexity of scent inheritance in hybrid roses. Spiller et al. [29] conducted a study to investigate the inheritance patterns of specific scent components in diploid rose genotypes. It was found that the scent components in hybrid progenies exhibited two different segregation rates. In a separate study by Nadeem et al. [30], hybridization experiments were performed using modern rose varieties with different scent profiles, and the resulting progenies exhibited varying degrees of scent intensity. Strong-scented progenies were obtained from crosses between roses with a strong scent, while crosses between roses with a moderate scent and roses with a strong scent resulted in progenies with a moderate scent. Combinations of strong × moderate scents resulted in progenies with a moderate scent, while combinations of strong × weak scents produced progenies with a weak scent.

It is evident from the studies that the scent trait possessed by parents is not observed in the progenies as expected, both in terms of intensity and compound composition, and obtaining scent progenies is challenging. Similarly, in this study, it was determined that the expected ratios between scentless × strongly scented combinations differed from each other. None of the genotypic predictions, according to the Punnett square, matched the phenotypic segregation. The rate of scented progeny in all combinations was much lower than expected, regardless of scent intensity, and it was difficult to obtain scented progeny. All these findings indicate that the scent trait may be controlled by polygenic genes, and the genes responsible for its inheritance may be recessive. It may also suggest that recessive alleles could be more prevalent than dominant ones. Since breeding studies have been focused more intensively on characteristics such as flower shape, flower stem length, and vase life, obtaining scented varieties with long vase lives using conventional breeding methods has been challenging. Consequently, scent has not been considered a prioritized selection criterion for many years [11,31,32,33]. This may be attributed to the elimination of ethylene-sensitive progeny to enhance postharvest durability and the negative selection of scented progenies due to their shorter postharvest.

### 2.4. Petal Number and Flower Doubleness

The number of petals varied between 6.67 and 155.56 among the F_1_ progenies, regardless of the cross combinations. The percentage of F_1_ progenies with single petals in all combinations was 1.63%, while the percentage of progenies with 41 or more petals was 21.14%. There was no F_1_ progeny in the single petal group in the cross combinations where Cabbage rose was used as the pollen parent. At least 50% of the F_1_ progenies obtained from all combinations, except for First Red × Black rose, fell into the full and very full groups. F_1_ progenies with more than 100 petal numbers were obtained from Layla × Black rose, Layla × Cabbage rose, and First Red × Cabbage rose. There was only one F_1_ progeny each in Samourai × Cabbage rose and Sweet Avalanche × Cabbage rose (Figure 3).

In studies on the inheritance of the number of petals and/or the flower doubleness trait in roses, it has been reported that the double flower trait may be controlled by a single dominant gene, while the single flower trait may be homozygous recessive [13,14,21]. Debener [21] conducted a crossbreeding study using various hybrid rose genotypes that possessed double flowers. As a result, progeny with single flowers was obtained. In the same study, it was observed that out of 109 hybrid progenies, 79 had double flowers (>7 petals) and 30 had single flowers (<7 petals). The number of petals in double-flowered genotypes ranged from 15 to 82, and it was noted that there was a decrease in the number of male organs with an increase in the number of petals. Shupert [22] stated that, as a result of hybridization between parents with single and double flowers on roses, 8 out of 19 hybrid progenies had double flowers, while 11 of them had single flowers. The researcher also stated that the parent with double flowers might be heterozygous in terms of flower form. Jones [14] created different cross combinations with single and double flowers in roses and obtained findings that the double flower trait was controlled together with the additive genes that determine the number of petals, except for two cross combinations. The researcher reported that the deviation in segregation rates in cross combinations that did not conform to this hypothesis was likely due to chance, with a probability of 30%. According to Nadeem et al. [30], as a result of morphological observations and measurements made on hybrid progenies obtained from 30 different combinations with 9 different hybrid rose varieties, it was determined that the number of petals varied between 16 and 40 according to the combinations and that some cross combinations showed positive heterosis and some showed negative heterosis.

In the results of this study, similar to the study by Debener [21], it was observed that the majority of F_1_ progenies exhibited the double flower trait, while a few progenies had single flowers. The production of progeny with single flowers, despite all parental genotypes being double-flowered (a range of 27 to 97 petals and being classified as full blooms or very full blooms according to ARS standards), suggests that the parents may be heterozygous for this trait, assuming that the trait is controlled by a single gene and the single-flower trait is homozygous recessive. However, the variation in segregation ratios among the combinations, the variation in the number of petals within the same progeny during both flowering periods, the presence of individuals with a lower number of petals than their parental genotypes, as well as individuals with significantly higher numbers of petals among the F_1_ progenies (the heterosis and heterobeltiosis values indicating that 65.12% of the progenies exhibited negative heterosis and 58.91% showed heterobeltiosis are provided in Appendix A), and the deviation of the observed phenotypic segregation ratios from the phenotypic distribution predicted by the Punnett square and the genotypic segregation ratios suggest gene interactions. Similarly, Jones [14] reported that two genes play a role in determining the flower form of roses. While one gene controls the double flower trait, the other additive gene influences the number of petals in genotypes exhibiting the double flower trait. However, the results of this study suggest that genetic interactions in petal number and flower doubleness traits exhibit a more complex mechanism than simple additive genetic effects. The presence of heterosis in the number of petals suggests the involvement of non-additive gene effects. Various studies have indicated a potential correlation between dominance, other non-additive genetic effects, and the formation of heterosis [34,35,36].

### 2.5. Flower Stem Length 

Among the F_1_ progenies, the flower stem length ranged from 6.30 cm to 87.20 cm. Notably, all F_1_ progenies with very short stem lengths were obtained from cross combinations using Black rose as the pollen parent. The majority of the F_1_ progenies exhibited flower stem lengths within the 30–49 cm range. The flower stem length of 3.84% of F_1_ progenies was shorter than 15 cm; 6.16% of them were between 16 cm and 29 cm; 80.39% of them were between 30 cm and 49 cm; 8.45% of them were between 50 cm and 69 cm; and 1.16% of them were over 70 cm (Figure 4).

Cut roses traded commercially are propagated through cuttings and/or grafting. Therefore, measuring the flower stem length in clonally propagated plants of the hybrid progeny can provide more reliable results regarding the known flower stem length of the progenies and their evaluation in terms of cut flower quality criteria. In breeding studies, measuring the flower stem length of F_1_ progenies propagated from seeds is considered to significantly contribute to the preliminary evaluation of the “short–medium–long” classification for the measured hybrid progeny’s flower stem length. Indeed, our F_1_ progenies, which were grown under the same conditions and clonally propagated, also exhibited flower stem lengths falling within the same class, albeit with varying range intervals.

The F_1_ progenies demonstrated variations in mid-parent and better parent heterosis. However, 96.51% of them displayed negative heterosis, indicating a performance decrease compared to the better parents, which were hybrid tea roses. Additionally, 58.91% of the progenies exhibited negative heterobeltiosis, indicating lower performance compared to the mid-parents (Appendix A). A significant percentage of the F_1_ progenies obtained in the study were classified as having a medium stem length in terms of flower stem length. Furthermore, the collected data revealed a continuous variation among the F_1_ progenies. When considering the overall average rates of the F_1_ progenies representing all cross combinations, it can be observed that they are distributed around a general mean in a manner consistent with a normal distribution curve. This, in conjunction with the presence of heterosis, supports Byrne’s [37] hypothesis that traits related to growth type, such as flower stem length, branch number, and plant height, are controlled by polygenic genes.

### 2.6. Flower Bud Length 

The average bud length of the F_1_ progenies obtained from different combinations was determined to be 3.42 cm, with bud lengths ranging from 1.48 cm to 5.97 cm depending on the specific cross combinations. The combination of Magnum × Cabbage rose produced the longest average bud length of 4.69 cm. Some combinations involving Black rose as the pollen parent resulted in buds smaller than 2 cm. Conversely, the Layla × Black rose combination produced F_1_ progenies with bud lengths approaching 6 cm. In terms of bud length, more than 50% of the F_1_ progenies were classified as large or very large (Figure 5).

Although bud length is a trait related to bud size and petal length in roses, limited research has been conducted specifically on the inheritance of bud length. However, there is existing information regarding the inheritance of petal length. It has been reported that petal length in roses is controlled by polygenic genes [38]. Shupert [22] investigated petal length in the F_1_ progeny obtained from crosses between *R. wichuraiana* ‘Basyes Thornless’ and *R. chinensis* ‘Old Blush,’ as well as in the F_2_ progeny resulting from backcrossing three hybrid progenies with *R. chinensis* ‘Old Blush.’ This study revealed wide variation in petal length among both the F_1_ and F_2_ progenies, with the average petal length of the F_1_ progenies being lower than that of the parents. This led to the conclusion that additive genes play a significant role in the inheritance of petal length. In another hybridization study conducted by Nadeem et al. [30] on modern rose varieties, petal lengths were examined, and it was observed that petal length exhibited positive heterosis in certain cross combinations while showing negative heterosis in others.

The findings obtained in this study suggest that, similar to petal length, bud length could also be controlled by polygenic genes. It was observed that the bud lengths of the F_1_ progeny exhibited negative transgressive segregation in some cross combinations. For example, in the Samourai × Black rose combination, while the average bud length of the parents was 4.65 cm (5.8 cm × 3.5 cm), the average bud length (3.36 cm) and the longest bud length (3.65 cm) in the F_1_ progeny remained below this value. Similarly, in the Magnum × Cabbage rose combination, the average bud length in the F_1_ progeny was 4.69 cm, whereas the average bud length of the parents was 5.1 cm (5.7 cm × 4.5 cm). In terms of mid-parent and better parent heterosis in the F_1_ progeny, 99.61% of the progenies exhibited negative heterosis, indicating a decrease in performance compared with the better parent. Furthermore, 92.25% of the progenies displayed negative heterobeltiosis, indicating a significant reduction in performance even when compared to the mid-parent (Appendix A).

### 2.7. Flower Diameters

The flower diameters of the F^1^ progenies ranged from 3.78 cm to 12.81 cm, with a mean diameter of 7.54 cm. When considering the different cross combinations, 4.62% of the F_1_ progenies had small flower diameters, 36.15% had medium diameters, 37.31% had large diameters, and 21.92% had very large diameters. It was observed that F_1_ progenies with small flower diameters only occurred in combinations where the Black rose was used as the pollen parent. On the other hand, all progenies resulting from combinations involving Sweet Avalanche and Avalanche as seed parents had flower diameters above 5.0 cm (Figure 6).

In studies conducted to determine the heritability of flower diameter in roses, it has been reported that the trait is significantly influenced by both the seed and pollen parents as well as gene interactions [39]. Furthermore, it has been found to be associated with petal length rather than petal number [14]. Dugo et al. [39] observed that the average flower diameter ranged between 2.05 and 5.72 cm in the F_1_ progeny, resulting from crosses with parents having an average flower diameter of 4.00 cm. The researchers noted that the flower diameter trait exhibited transgressive variation in both positive and negative directions. Jones [14] determined that the flower diameter varied between 2.00 and 6.90 cm in progeny, resulting from crosses between genotypes with flower diameters ranging from 2.00 to 5.00 cm. Nadeem et al. [30] reported that flower diameters ranged from 3.00 to 5.00 cm in progeny obtained from crosses between modern rose varieties with flower diameters ranging from 4.00 to 6.00 cm. Additionally, they found that flower diameters displayed positive heterosis in some cross combinations, while negative heterosis was observed in others.

In this study, the obtained values for the flower diameter showed a wide variation. Some combinations exhibited negative transgressive segregations in terms of the flower diameter. For example, in the Layla × Black rose combination, the average flower diameter of the parents was 9.08 cm (10.15 cm × 8.00 cm), while the average flower diameter of the F_1_ progeny was determined to be 7.08 cm. In the Magnum × Cabbage rose combination, the average flower diameter of the parents was 11.08 cm (10.50 cm × 11.65 cm), while the average flower diameter of the F_1_ progeny was determined to be 10.15 cm. Additionally, the Black rose species, being the parent with the smallest flower diameter after the Damask rose species, produced progeny with a small flower diameter that was not observed in other pollen parents when used as pollen parents in combinations (Samourai × Black rose, Avalanche × Black rose). It has been determined that 94.19% of the progenies exhibit negative heterosis, and 84.88% exhibit negative heterobeltiosis (Appendix A). These findings support the quantitative inheritance of the flower diameter trait, as reported by other researchers, and indicate that some of the progenies fall below the mid-parents, making them unsuitable for certain market quality criteria for cut flowers.

### 2.8. Hierarchical Clustering Heat Map of Examined Traits 

A hierarchical clustering heat map of the average values of the F_1_ seedlings obtained from 10 different combinations and parents’ traits was plotted in order to determine the phenotypic relationship among parents and progenies. Based on the six characteristics examined, two main groups were formed. Seed parents differed from both pollen parents and F_1_ progenies in terms of flower diameter, bud length, and flower stem length. While the Cabbage rose was separated from the seed parents in terms of scent, petal number, and flower stem length, it differed from both pollen parents and F_1_ progenies in terms of all other traits except flower stem length. Except for the scent, the seedlings in all combinations where Black rose was the pollen parent showed similarities to Black rose. Seedlings obtained from combinations where Cabbage rose was the pollen parent were distinguished from F_1_ seedlings obtained from other combinations in terms of the petal numbers, except for Magnum × Cabbage rose. Flower diameter and bud length traits were included in the same subgroup and were more closely related to flower stem length than other traits. Similarly, the petal number and scent traits were included in the same subset and seemed to be more closely related to each other than other traits. Parents and combinations were clustered into two main groups. The first main group consisted of only seed parents. In the second main group, there were three different subgroups. The first group consisted only of Cabbage roses. The second group consisted of combinations where Cabbage rose was the pollen parent. The third group consisted of combinations where Black rose and Black rose were pollen parents. It is understood that F_1_ seedlings were more similar to pollen parents than seed parents in terms of the examined characteristics except for scent. In other words, hybrid seedlings obtained from hybridizations with old garden roses were generally of lower quality than commercially available modern rose varieties (Figure 7).

### 2.9. Variance Components, Heritability Estimation, and Phenotypic Correlation Matrix of Quantitative Traits

The values of variance components for the pollen parent (*σ*^2^*_α_*), seed parent within the pollen parent (*σ*^2^*_β_*_(*α*)_), error (*σ*^2^*_e_*), and narrow-sense heritability (*d*) estimates obtained for traits including flower stem length, number of petals, flower diameter, and bud length in roses obtained using Bayesian methods are shown in Table 1. The TAD (Trace–Autocorrelation–Density) plots are provided in Appendix A. The *σ*^2^*_α_* was determined to be higher than the *σ*^2^*_β_*_(*α*)_ for all the traits examined, and the *σ*^2^*_α_* was found to be 1.4 times greater than the *σ*^2^*_β_*_(*α*)_ in terms of flower stem length, 1.3 times greater in terms of petal number, and 2.0 times greater in terms of flower diameter and flower bud length. The *σ*^2^*_e_* was low for all of the variables (<3%), with the exception of flower stem length. Moreover, all quantitative traits had moderately low narrow-sense heritability. The flower diameter trait was determined to have the highest heritability, with a value of 46.9%, while flower stem length was identified as the trait with the lowest heritability, at 24.9%.

The phenotypic correlation matrix among flower stem length, petal number, flower diameter, and flower bud length revealed the following correlations: there was a weak positive correlation between flower stem length and flower diameter (r = 0.231) as well as flower bud length (r = 0.278). Furthermore, a strong positive correlation was observed between flower diameter and flower bud length (r = 0.904). However, no significant correlation was found between petal number and other traits (Table 2, Appendix A).

There are other studies in which the heritability of quantitative traits in roses is estimated. Panwar et al. [40] reported high heritability (>80%) for flower diameter, petal number, and plant height traits in roses. Gitonga [41] examined the broad-sense heritability of flower stem length and petal number in 148 F_1_ progenies obtained from the hybridization of tetraploid P540 and P867 rose genotypes. The heritability values for flower stem length ranged between 0.86 and 0.91, while the heritability values for petal number varied between 0.88 and 0.99. Environmental conditions were found to have minimal effects on these traits. In a study by Liang [42], the narrow-sense heritability for flower diameter was determined as 0.24, and the narrow-sense heritability for petal number was determined as 0.12. The researcher concluded that flower size traits were moderately narrow and highly broad heritable. Lau et al. [43] investigated narrow-sense and broad-sense heritability in diploid roses across different seasons. They found narrow-sense heritability for flower diameter to be 0.38 and broad-sense heritability to be 0.70. The narrow-sense heritability for petal number ranged between 0.26 and 0.33, and the broad-sense heritability ranged between 0.85 and 0.91 in two different years. The researchers emphasized that flower diameter was influenced not only by genetic factors but also by environmental effects. Soujanya et al. [44] examined heritability and genetic variability in 25 different hybrid tea roses and observed high heritability for all traits, including plant height. Wu et al. [45] reported a narrow-sense heritability of 0.50 for plant height in their study on diploid roses, indicating it to be a highly heritable trait.

In this study, the narrow-sense heritability of the flower stem length of 258 rose progenies was estimated to be 0.25, petal number 0.31, flower diameter 0.47, and bud length 0.37. Broad-sense heritability was not evaluated. Since all measurements were made under controlled conditions, there was no seasonal component or contribution to the variance. When comparing these results with the aforementioned studies, the heritability of petal number and flower diameter generally aligns but does not exactly overlap. There are limited studies on flower stem length, but the heritability reported by Wu et al. [45] partially corresponds to the findings of this study. No data were found regarding bud length. When comparing heritability, it is important to consider studies conducted on the same population and traits. Different calculation methods and environmental conditions may result in variations in narrow-sense heritability, especially for polygenic traits, due to the influence of environmental factors on genetic factors [46]. Therefore, a direct comparison should not be expected. In fact, this study used the Bayesian method [47], which is considered superior to the commonly preferred ANOVA and REML methods used in previous studies. Additionally, the population and climatic conditions differed from those of other studies.

In the study, the narrow-sense heritability of 0.47 calculated for the flower diameter trait was higher compared to the other traits. This indicates that the additive gene effect in the inheritance of the flower diameter is stronger than for other traits. However, the narrow-sense heritability of 0.25 calculated for the flower stem length trait was lower than that of the other traits. This suggests that the inheritance of the flower stem length is more influenced by non-additive gene effects compared to the other traits. Moreover, the narrow-sense heritability of the petal number and bud length showed similar values. The moderately narrow sense of heritability for all traits suggests the presence of non-additive gene effects on these traits. These results show that the inheritance of traits has a complex structure and that the effects of environmental factors should be evaluated.

Estimates obtained using the Punnett square and heritability estimations generally align with each other and partially correspond to the results reported in previous studies. Previous studies have suggested that traits such as flower doubleness and petal number may be regulated by a single dominant gene or exhibit an additive gene effect. However, both the Punnett square analysis and narrow-sense heritability estimates in this study indicate the presence of both additive and non-additive gene effects. The findings of Gitonga et al. [41] also support the possibility of polygenic control for these traits. The moderate, narrow-sense heritability estimates align with the notion that flower diameter, flower stem length, and bud length traits may be influenced by polygenic genes. The non-additive gene effect may involve one parent’s genes dominating those of the other parent, and the observation that all progenies display dominant traits inherited from the pollen parent can serve as evidence of the non-additive gene effect on these traits.

Narrow-sense heritability can serve as an indicator of the rate at which a desired trait can progress through selection [48], and the magnitude of heritable variability is the most significant factor in terms of its impact on the response to selection [49]. In this study, the moderately narrow sense heritability values obtained for all traits suggest that selections with moderate intensity for flower diameter, petal number, and bud length can be made under controlled conditions during the third flowering period. However, performing selection during later flowering periods would be more appropriate under field conditions. Furthermore, for all traits, including flower stem length, it would be more suitable to conduct a specific selection for each season, and multiple measurements should be taken for each progeny. It is predicted that a single measurement would not be sufficient to fully characterize the genotype. In fact, Kawamura et al. [50] also noted that the non-genetic variance, along with differences between plants, is significantly influenced by variations in measurements taken on the same plant.

When examining the phenotypic correlation matrix, a significantly high positive genetic relationship was observed between flower diameter and bud length, while no significant relationship was found between petal number and flower diameter (*p* < 0.01). These results align with the findings of Jones [14], who also reported no correlation between flower diameter and petal number. Jones suggested that flower diameter was more related to petal length than to petal number. The exact relationship between petal and bud lengths is not yet known. However, in some roses, the outer row of petals is longer than the inner row of petals. Therefore, if bud length is solely evaluated in relation to the outer row of petals, the relationship may be misinterpreted. Akhtar et al. [51] determined a relationship between flower diameter and bud length, suggesting that these traits may be controlled by common or related genes, while environmental factors that influence both traits may also play a role. The weak but positive relationship between flower stem length and flower diameter, as well as bud length, may indicate that stem length has some effect on flower size. However, there are likely other factors that have a more significant impact on flower size than stem length. Indeed, Plaut et al. [52] reported that flower size varies not only based on flower stem length but also on the number and size of leaves on the flower stem.

## 3. Materials and Methods

In this study conducted between 2018 and 2020, F_1_ progeny cultivation and the measurement of various quantitative and qualitative traits were carried out in a modern plastic greenhouse at the Department of Horticulture, Faculty of Agriculture, Ankara University, located in Ankara province, Turkey [39°57′53.8″ N, 32°51′50.8″ E, according to Google Maps, 2020] [53].

### 3.1. Plant Material

The parent plants used in the study titled “Success of Hybridization in Hybrid Tea Rose × Old Garden Rose Combinations” and the progeny resulting from the hybridization of these parents were used as plant materials. Some qualitative and quantitative traits of parents are given in Table 3.

A total of 18 combinations were derived from the F_1_ progenies, where Layla, Magnum, Sweet Avalanche, Samourai, and Avalanche were used as the seed parents and Black rose, Cabbage rose, and Damask rose were used as the pollen parents.

The process from the procurement of plant material (including planting, applied cultural practices, determination of ploidy levels and pollen quality, pollination studies, seed collection, cold stratification, and seed sowing) to transplanting the F_1_ progenies is extensively described in the study titled “Success of Hybridization in Hybrid Tea Rose × Old Garden Rose Combinations”.

### 3.2. Transplanting and Cultural Practice of F_1_ Seeds

After the F_1_ progenies developed their first four true leaves, they were transplanted into pots filled with a 3:1 *v*/*v* mixture of coco peat and pumice. In April 2019, measurements were taken to assess the morphological characteristics of the F_1_ progenies. Irrigation, fertilization, and other cultural practices applied to the F_1_ progenies were carried out similarly to those used for the parent plants.

### 3.3. Qualitative and Quantitative Traits Examined in F_1_ Progenies

During two flowering periods in one-year-old F_1_ progenies, several characteristics were examined under the same conditions. The details of the traits examined are provided below:

Recurrent blooming: The F_1_ progenies were categorized based on their flowering behavior. Those that bloomed more than once a year were classified as having recurrent blooming (+), while others were indicated as having no recurrent blooming (−).

Scent: The scent of F_1_ progenies was evaluated using the magnitude estimation procedure of the sensory evaluation method [54]. The classification process was conducted by the author team and relied on their personal experiences and perceptions. A more intuitive approach was used to assess scent sensitivity. The rose species known for their strong scent, namely Black rose, Damask rose, and Cabbage rose, which were also used as parents in this study, were used as reference points, and the scent intensity was considered 4 points. Scentless commercial hybrid varieties were rated at 1 point. Intermediate classes are relatively established. The results obtained at each flowering period were checked for consistency. Scent intensity was evaluated in a well-ventilated and isolated room during the morning hours when the flowers were in full bloom. Scent intensity was divided into four classes: scentless and barely perceptible (1 point), slightly scented (2 points), moderately scented (3 points), and strongly scented (4 points).

Petal number and flower doubleness: When the stigma and anthers were visible in the flowers of F_1_ progenies, the number of petals was counted for at least two flowers in each genotype [55]. Only the petals of terminal flowers were considered. The F_1_ progenies were classified according to the criteria set by the American Rose Society: single (4–8 petals), semi-double (9–16 petals), double (17–25 petals), full (26–40 petals), and very full (≥41 petals). These criteria were considered more comprehensive than those of UPOV and more suitable than commercial quality criteria [56].

Flower stem length: The length of the flower stem was measured in cm for flowers harvested from the bottom of the second 5-part leaf [57]. The distance from the cut point to the apex of the bud was recorded. The flower stem length of genotypes was divided into five classes: very short (≤15 cm), short (16 to 29 cm), medium (30 to 49 cm), long (50 to 69 cm), and very long (≥70 cm) based on the obtained measurements.

Flower bud length: The length of flower buds in mature F_1_ progenies was determined by measuring the distance between the lower and upper ends of the bud using a digital caliper. Only terminal flowers were considered for measurement. The bud length of genotypes was categorized into four classes: small (1.2 to 2.5 cm), medium (2.5 to 3.0 cm), large (3.1 to 4.9 cm), and very large (≥5.0 cm) based on the obtained measurements.

Flower diameter: Fully blooming flowers were harvested, and their diameters were measured using a digital caliper. Measurements were taken on terminal flowers. The flower diameters of genotypes were divided into four classes: small (3.0 to 4.9 cm), medium (5.0 to 6.9 cm), large (7.0 to 8.9 cm), and very large (≥9.0 cm), based on the obtained measurements and following the UPOV guidelines [58].

### 3.4. Data Analysis

The Punnett square was used to theoretically predict the inheritance of qualitatively measured traits such as recurrent blooming, scent, and flower doubleness. Heterosis and heterobeltiosis were calculated for quantitative traits such as petal number, flower diameter, flower stem length, and bud length, following the formula described by Nadeem et al. [30]: $$Ht\%=\left[\frac{F_{1}-BP}{BP}\right]\times 100\;\;\;\;\;Hbt\%=\left[\frac{F_{1}-MP}{MP}\right]\times 100$$Ht = heterosis, BP = better parent value, Hbt = heterobeltiosis, MP: mid-parent value.

In conjunction with heterosis and heterobeltiosis, the Bayesian variance component estimation method was employed to estimate the heritability and phenotypic correlation of these quantitative traits. For the analysis of the two-way random nested design using the Bayesian method, the model provided by Kaya Başar and Fırat [47] was used. To perform a Bayesian analysis, both the likelihood function based on the observed data and the prior distributions for each parameter of the model are needed, as they form the components of the posterior distribution. The Gibbs sampling algorithm was used to apply Bayesian methods. The Gibbs sampling algorithm is a method for generating random samples from the full conditional distributions of the parameters without having to calculate the density. It starts with an initial starting point for the parameters and samples a value for each parameter of interest one at a time, given the values for the other parameters and data. Once all of the parameters of interest have been sampled, the nuisance parameters are sampled, given the parameters of interest and the observed data. This process is then repeated. The power of the Gibbs sampling algorithm is that it allows the joint distribution of the parameters to converge to the joint probability of the parameters given the observed data. After estimating the variance components of the pollen parent (*σ*^2^*_α_*), seed parent within pollen parent (*σ*^2^*_β_*_(*α*)_), and error (*σ*^2^*_e_*), as well as the phenotypic correlations, the heritability coefficient (*d*) was estimated by using the variance components. The statistical analysis for the Bayesian method was performed using the BGLIMM procedure within the SAS software version 9.4. Moreover, a heat map was generated to visualize the hierarchical clustering, and the values were standardized for each combination to evaluate the relationships between the examined quantitative and qualitative traits.

## 4. Conclusions

In this study, the obtained F_1_ progenies deviated from the commercial quality criteria as cut flowers and exhibited traits more similar to those of old garden roses. The moderately narrow sense of heritability observed in the examined traits indicated the influence of non-additive gene effects. In terms of flower diameter, bud length, and flower stem length traits, it appears that old garden roses may exert a dominant effect on hybrid tea roses when used as pollen parents. However, this was not the case for the scent trait. The similarity of the F_1_ progenies to the seed parents in terms of scent suggests that the scent trait may be more dominant in hybrid tea roses than in old garden roses. The inheritance of the scent trait seems to be particularly challenging due to the complex genetic background of roses, necessitating the use of molecular methods to elucidate it. When evaluating these combinations, especially for purposes such as disease resistance and stress tolerance, it may be more appropriate to include combinations where old garden roses are used as pollen parents to achieve the desired outcomes in breeding programs. It can be a challenge for cut flower breeding programs.

During the rose selection, it is advisable to avoid a rigorous selection based solely on the first two flowering periods. Instead, repeated measurements should be taken for each plant, and if necessary, separate measurements should be made for each season. Although flower doubleness is clearly understood in the first flowering period, the variation in the petal number can sometimes be misleading. For selections based on flower diameter and bud length, evaluating either one of these traits alone may be sufficient.

The results of this study are predicted to guide interspecies hybridization in roses and contribute to breeding programs aiming to obtain diverse hybrid populations. By using available data on parental performance, the selection of cross combinations can be more targeted, increasing the likelihood of success compared with random selection. Moreover, the selection of parents and identification of cross combinations entail significant costs, efforts, and time. Expanding the seed and pollen parent gene pool makes a great contribution and is highly recommended for breeders. While large-scale hybridization strategies employed by well-funded international breeding programs may not be feasible for many public-sector national breeding programs, adopting fewer but carefully chosen cross combinations and parent selection methods can enhance the efficiency of breeding programs, leading to the introduction of new varieties into the market. It is anticipated that these findings will contribute significantly to genotype selection during parental and progeny selection. Subsequent steps in this study should involve investigating the contribution of old garden roses as seed parents to the inheritance of traits. Further cross-studies on trait inheritance need to be conducted, with particular emphasis on incorporating molecular approaches, especially for complex traits such as scent that are challenging to inherit.

## Figures and Tables

**Figure 1 plants-13-01797-f001:**
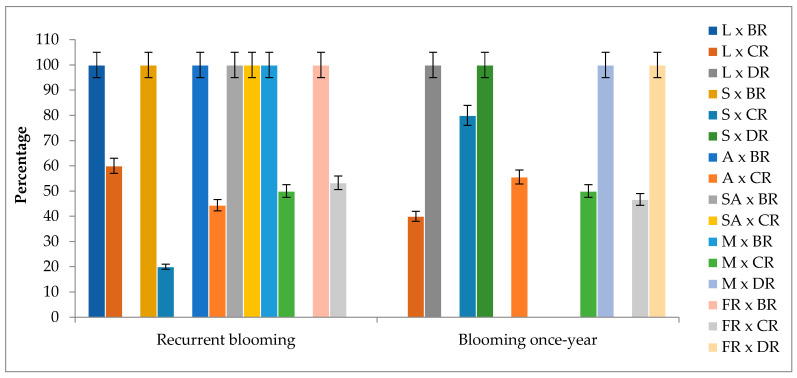
Recurrent blooming rate of F_1_ progenies according to cross combinations. L: Layla, S: Samourai, A: Avalanche, SA: Sweet Avalanche, M: Magnum, FR: First Red, BR: Black rose, CR: Cabbage rose, DR: Damask rose; error bars show error percentage (5%).

**Figure 2 plants-13-01797-f002:**
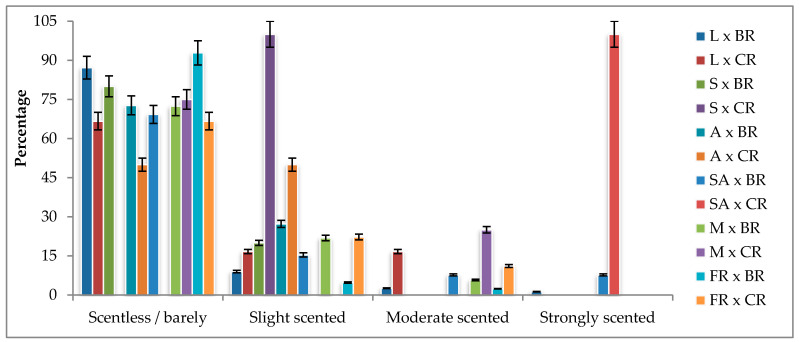
Scent classes of F_1_ progenies according to cross combinations (%). L: Layla, S: Samourai, A: Avalanche, SA: Sweet Avalanche, M: Magnum, FR: First Red, BR: Black rose, CR: Cabbage rose; error bars show error percentage (5%).

**Figure 3 plants-13-01797-f003:**
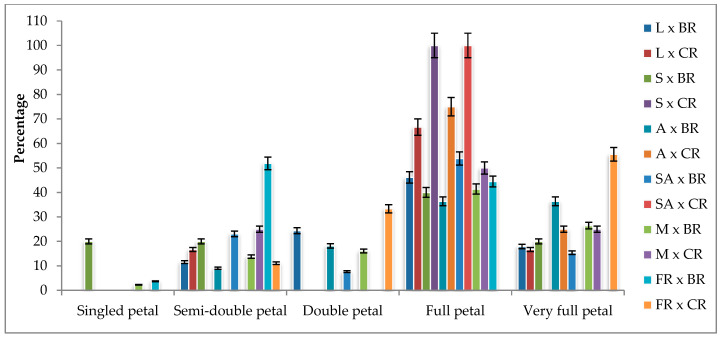
Petal number classes of F_1_ progenies according to cross combinations (%). L: Layla, S: Samourai, A: Avalanche, SA: Sweet Avalanche, M: Magnum, FR: First Red, BR: Black rose, CR: Cabbage rose; error bars show error percentage (5%).

**Figure 4 plants-13-01797-f004:**
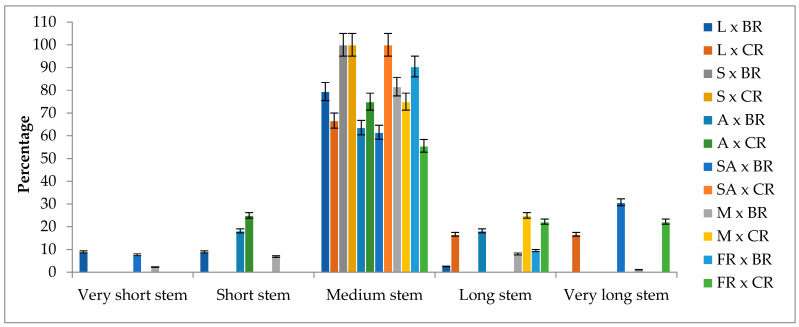
The flower stem length classes of F_1_ progenies according to cross combinations (%). L: Layla, S: Samourai, A: Avalanche, SA: Sweet Avalanche, M: Magnum, FR: First Red, BR: Black rose, CR: Cabbage rose; error bars show error percentage (5%).

**Figure 5 plants-13-01797-f005:**
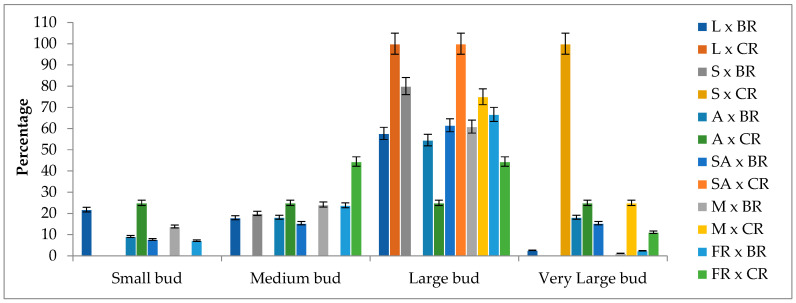
Flower bud length classes of F_1_ progenies according to cross combinations (%). L: Layla, S: Samourai, A: Avalanche, SA: Sweet Avalanche, M: Magnum, FR: First Red, BR: Black rose, CR: Cabbage rose, error bars show error percentage (5%).

**Figure 6 plants-13-01797-f006:**
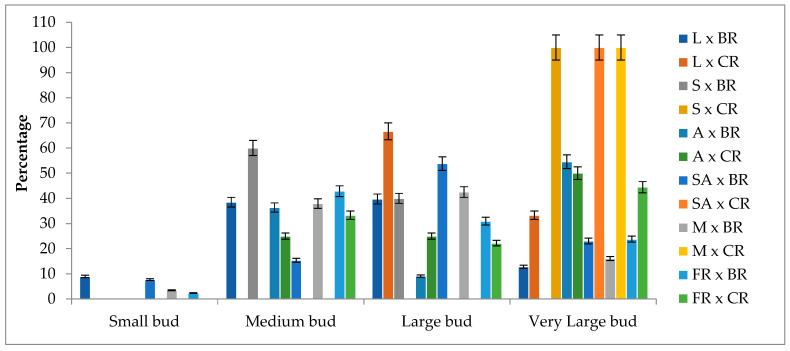
Flower diameter classes of F_1_ progenies according to cross combinations (%). L: Layla, S: Samourai, A: Avalanche, SA: Sweet Avalanche, M: Magnum, FR: First Red, BR: Black rose, CR: Cabbage rose, error bars show error percentage (5%).

**Figure 7 plants-13-01797-f007:**
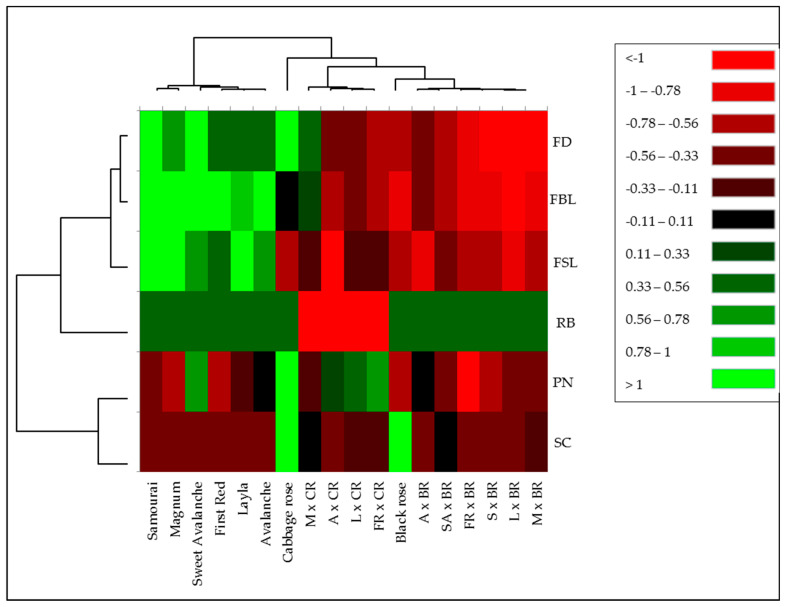
Hierarchical clustering heat map of examined traits and cross combinations. The color-coded scale indicates an increase from red to black and then green. M: Magnum, A: Avalanche, L: Layla, FR: First Red, SA: Sweet Avalanche, S: Samourai, BR: Black rose, CR: Cabbage rose, FD: flower diameter, FBL: flower bud length, FSL: flower stem length, RB: recurrent blooming, PN: petal number, SC: scent.

**Table 1 plants-13-01797-t001:** Bayesian estimation of variance components and heritability coefficient based on variance components from the pollen parent.

Quantitative Traits	*σ* ^2^ * _α_ *	*σ* ^2^ * _β_ * _(_ * _α_ * _)_	*σ* ^2^ * _e_ *	*d*
Flower stem length	11.506	7.795	135.60	0.25
Petal number *	0.027	0.021	0.2558	0.31
Flower diameter	0.588	0.292	2.8707	0.47
Flower bud length	0.090	0.045	0.6011	0.37

* The data were normalized on a logarithmic scale due to the non-normal distribution of the dataset (Appendix A).

**Table 2 plants-13-01797-t002:** Phenotypic correlation matrix among quantitative traits.

Quantitative Traits	Flower Stem Length	Petal Number	Flower Diameter	Flower Bud Length
Flower stem length	1.00			
Petal number	0.069	1.00		
Flower diameter	0.231 **	0.070	1.00	
Flower bud length	0.278 **	0.077	0.904 **	1.00

** *p* ≤ 0.01.

**Table 3 plants-13-01797-t003:** Some qualitative and quantitative traits of rose genotypes were used as parents.

Genotypes	Recurrent Blooming	Scent	Petal Number (pcs/per Flower)	Flower Stem Length (cm)	Flower Bud Length (cm)	Flower Diameter (cm)
Layla	year-round	Scentless	34/full blooms	70.0	5.4	10.15
Samourai	year-round	Scentless	33/full blooms	85.0	5.8	12.40
Avalanche	year-round	Scentless	40/full blooms	65.0	5.9	10.05
Sweet Avalanche	year-round	Scentless	51/very full	65.0	6.4	11.90
Magnum	year-round	Scentless	28/full blooms	90.0	5.7	10.50
First Red	year-round	Scentless	29/full blooms	60.0	5.5	10.10
Black rose	year-round	Strong	27/full blooms	40.0	3.5	8.00
Cabbage rose	year-round	Strong	97/very full	40.0	4.5	11.65
Damask rose	once a year	Strong	30/full blooms	40.0	2.5	6.02

All traits of the parents were determined using the method described in “Morphological Characteristics Examined in F_1_ Progenies”.

## Data Availability

The original contributions presented in the study are included in the article/Appendix A.

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
