# Peer review of "Inheritance of Some Traits in Crosses between Hybrid Tea Roses and Old Garden Roses"

_plants, 2024, doi:10.3390/plants13131797_

Round 1

Reviewer 1 Report

Comments and Suggestions for Authors

In order to explore the genetic mechanism of rose traits and develop new aroma varieties that meet the quality standards of commercial rose varieties. This study investigated and analyzed multiple traits of hybrid tea roses, old garden roses, and their hybrid offspring. This includes repeated flowering, flower stem length, flower diameter, number of petals, flowering odor, and flower buds, aiming to evaluate the ability of old garden roses to transfer certain plant and flower shapes to the next generation. By collecting trait information, it was found that compared to hybrid roses, the offspring traits are more similar to those of old garden roses. Meanwhile, studying the inheritance of odor traits can help cultivate new high-quality rose materials for fragrance varieties. This work has certain scientific value and may be published after proposing some revisions. Some suggestions are as follows.

1. Please maintain consistent font in all images.

2. Please indicate significance in the bar chart.

3. Use sensory evaluation methods to identify odor characteristics, whether there are artificial differences, and how to eliminate differences. What is the sample size and is the sensitivity of each participant to the same level of odor consistent.

4. Determine whether the F1 generation is genetically stable and why not use F3 to represent the traits of the phenotype.

5. "L X DR" is included in Figure 1, but not in Figure 2 and others, please explain the reasons why each combination method is different.

6. Please elaborate on the representative characteristics of the experimental materials selected in this study.

Author Response

Reviewer 1

In order to explore the genetic mechanism of rose traits and develop new aroma varieties that meet the quality standards of commercial rose varieties. This study investigated and analyzed multiple traits of hybrid tea roses, old garden roses, and their hybrid offspring. This includes repeated flowering, flower stem length, flower diameter, number of petals, flowering odor, and flower buds, aiming to evaluate the ability of old garden roses to transfer certain plant and flower shapes to the next generation. By collecting trait information, it was found that compared to hybrid roses, the offspring traits are more similar to those of old garden roses. Meanwhile, studying the inheritance of odor traits can help cultivate new high-quality rose materials for fragrance varieties. This work has certain scientific value and may be published after proposing some revisions. Some suggestions are as follows.

  1. Please maintain consistent font in all images.
  2. Please indicate significance in the bar chart.
  3. Use sensory evaluation methods to identify odor characteristics, whether there are artificial differences, and how to eliminate differences. What is the sample size and is the sensitivity of each participant to the same level of odor consistent.
  4. Determine whether the F1 generation is genetically stable and why not use F3 to represent the traits of the phenotype.
  5. "L X DR" is included in Figure 1, but not in Figure 2 and others, please explain the reasons why each combination method is different.
  6. Please elaborate on the representative characteristics of the experimental materials selected in this study.

Authors:

  1. A consistent font has been used in all images.
  2. The significance of error bars has been indicated in all bar charts.
  3. Our scent classification is primarily a subjective assessment. The classification process is based on the personal experiences and perceptions of our team of authors, and we have taken a more intuitive approach to assessing scent sensitivity. Black rose, Damask rose and Cabbage rose are widely known and recognized for their strong scents. Hybrids that emitted a scent intensity equal to or greater than these roses were considered to have a strong scent based on the parental scent. The main factor in keeping the classification broad is the variability in scent perception. It is unrealistic to claim that an intensely scented rose is scentless (it may be imperceptible to the human nose). At this point, a reliable distinction was made between the two extreme classes (strongly scented, scentless). The intermediate classes are relatively well defined. Moreover, this process is supported by numerous experiences and observations. The results obtained in each flowering period are consistent. Furthermore, the scent trait was consistently monitored by the same four people in every flowering period except for the 2018-2020 period when the study was conducted. We therefore consider the results reliable. We believe that this approach is different from traditional methods of scent classification, but we consider it sufficient to validate our results. However, if deemed insufficient by the reviewer, we are willing to remove the scent feature from the manuscript.

The text has also been corrected as follows:

‘Scent: The scent of F1 progenies was evaluated using the magnitude estimation procedure of sensory evaluation method [54]. The classification process was conducted by the author team and relied on their personal experiences and perceptions. A more intuitive approach was used to assess scent sensitivity. The rose species known for their strong scent, namely Black rose, Damask rose, and Cabbage rose, which were also used as parents in this study, were used as reference points, and the scent intensity was considered as 4 points. Scentless commercial hybrid varieties were rated as 1 point. Intermediate classes are relatively established. The results obtained at each flowering period were checked for consistency. Scent intensity was evaluated in a well-ventilated and isolated room during the morning hours when the flowers were in full bloom. Scent intensity was divided into four classes: scentless and barely perceptible (1 point), slightly scented (2 points), moderately scented (3 points), and strongly scented (4 points).’

  1. As with many other woody fruit species in the Rosaceae family, roses are susceptible to cross-pollination and contain a high degree of heterozygosity. For this reason, a wide variation could be obtained in the F1 generation from the hybrid tea roses and old garden roses hybridization, which have two different origins. The hybrid offspring determined by selection from this variation in the F1 generation will be propagated using vegetative methods and new variety candidates will be obtained. When it is considered to obtain F2 and F3 generations by selfing F1 plants, the superior agronomic qualities of the selected F1 plants will not be maintained since genetic expansion will occur. Since new variants will emerge in each selfing generation, the selection stage will have to be carried out in the later progeny stages. In addition, The F3 generation may contain more genetic variation than the F1 generation, which may increase the likelihood of undesirable phenotypic traits. Especially in vegetatively propagated plants such as as roses, these genetic variations can lead to undesirable results and reduce commercial value. It is thought that F1 generation plants that can be propagated using vegetative propagation methods such as grafting eyes, cuttings and in vitro micropropagation may be sufficient to obtain new rose variety candidates. It is not foreseen that a factor such as the possible occurrence of a mutation, which may lead to a deterioration of genetic stability, may cause a difference between F1 plants and F3 plants. For plant species propagated by seed, selection at later progeny stages such as F2 and F3 increases the possibility of utilizing a wider variation. However, as in Rosa sp., sufficient variation may occur in F1 progeny obtained from the hybridization of two different heterozygous genetic structures. For this reason, qualified hybrid offspring were selected at the F1 generation stage to form elite rootstock (not rootstock rootstock, but vegetative starting material as a source) material for vegetative propagation. It was considered sufficient to carry out the observations at the F1 stage.
  2. The combination 'L X DR' did not yield flowers due to an extended period of juvenile sterility. Therefore, the results for this combination were not included in the graphs examining flower characteristics. This is explained under the heading '2.1. Qualitative And Quantitative Traits of F1 Progenies' within the manuscript.
  3. It has been added as supplementary material.

Reviewer 2 Report

Comments and Suggestions for Authors

Very interesting work, however in my opinion there are some unclear things and some things to add.

Introduction: In the introduction it would be interesting to include the number of species belonging to the Rosa genus. Also enter the genetic aspects such as ploidy.

Materials and methods: In the materials and methods the number of plants, for each crossing combination, in which the traits were detected is unclear.

- In figure 1 there are 12 crossing combinations and 16 columns in the graph. Why?

 - Also check the other figures as there does not seem to be a correspondence between the number of the crossing combinations and the columns in the graph relating to the traits

In lines 109,116,139,163,172,175,231,320,354,360,440,449,455,456,465,491,507,518,524 end 599 et al. in italic

Author Response

Reviewer 2

Very interesting work, however in my opinion there are some unclear things and some things to add.

  1. Introduction: In the introduction it would be interesting to include the number of species belonging to the Rosa genus. Also enter the genetic aspects such as ploidy.
  2. Materials and methods: In the materials and methods the number of plants, for each crossing combination, in which the traits were detected is unclear.
  3. In figure 1 there are 12 crossing combinations and 16 columns in the graph. Why? Also check the other figures as there does not seem to be a correspondence between the number of the crossing combinations and the columns in the graph relating to the traits
  4. In lines 109, 116, 139, 163, 172, 175, 231, 320, 354, 360 ,440, 449 ,455 ,456, 465, 491, 507, 518, 524 and 599 et al. in italic

Authors:

  1. The following statement has been added as the first paragraph of the introduction section:

‘The rose, which belongs to the Rosa genus in the Rosaceae family, is naturally dis-tributed in the Northern Hemisphere, including Asia, Europe, the Middle East, and North America. It is one of the plant species widely used in the ornamental plant industry, cosmetic industry, food, and medicine sectors [1]. There are more than 100 to 250 species of roses identified worldwide [2]. The haploid chromosome number of roses is x=7, and chromosome numbers vary depending on the ploidy level, ranging from 2n=2x=14 in diploids to 2n=8x=56 in octoploids. In a recent study, an endemic wild rose species (Rosa praelucens Byhouwer) with a chromosome number of 2n=10x=70 has also been reported [3]’.

It has been rearranged according to the additional sources added in the references.

  1. The number of F1 individuals varies according to the success of the combinations. To avoid confusion, the numbers in the combinations have not been specified in the main text. However, as per another reviewer's suggestion, all the characteristics of the F1 individuals have been provided as supplementary material. The number of individuals per combination is clearly indicated in this file.

When evaluating the numbers of hybrid individuals in combinations, it is important to note the occurrence of embryo abortion in roses, which means that the desired and expected number of hybrid individuals may not be obtained. This already emphasizes the importance of combination success.

  1. The graph area has become narrow, and not all combinations are visible in the graph area. By expanding the graph area, all combinations have been made visible. Taking Graph 1 as an example, there are 16 columns corresponding to 16 combinations. All graphs have been checked.
  2. Unfortunately, we were unable to implement this correction as the use of 'et al.' in italics is not specified in the guidelines for article writing. The writing format of 'et al.' is demonstrated on the last line of the first page in the PDF file opened from the provided link: 'https://mdpi-res.com/data/mdpi_references_guide_v9.pdf'.

Round 2

Reviewer 1 Report

Comments and Suggestions for Authors

The questions raised have been revised.